# Few-shot Relational Reasoning via Connection Subgraph Pretraining

**Qian Huang**[*]
Stanford University
qhwang@cs.stanford.edu

**Hongyu Ren**[*]
Stanford University
hyren@cs.stanford.edu

**Jure Leskovec**
Stanford University
jure@cs.stanford.edu

## Abstract

Few-shot knowledge graph (KG) completion task aims to perform inductive reasoning over the KG: given only a few *support* triplets of a new relation ⋈ (*e.g.*, (chop, ⋈, kitchen), (read, ⋈, library)), the goal is to predict the *query* triplets of the same unseen relation ⋈, *e.g.*, (sleep, ⋈, ?). Current approaches cast the problem in a meta-learning framework, where the model needs to be first jointly trained over many *training few-shot tasks*, each being defined by its own relation, so that learning/prediction on the *target few-shot task* can be effective. However, in real-world KGs, curating many training tasks is a challenging *ad hoc* process. Here we propose Connection Subgraph Reasoner (CSR), which can make predictions for the target few-shot task directly without the need for pre-training on the human curated set of training tasks. The key to CSR is that we explicitly model a shared *connection subgraph* between *support* and *query* triplets, as inspired by the principle of eliminative induction. To adapt to specific KG, we design a corresponding self-supervised pretraining scheme with the objective of reconstructing automatically sampled connection subgraphs. Our pretrained model can then be directly applied to target few-shot tasks on without the need for training few-shot tasks. Extensive experiments on real KGs, including NELL, FB15K-237, and ConceptNet, demonstrate the effectiveness of our framework: we show that even a learning-free implementation of CSR can already perform competitively to existing methods on target few-shot tasks; with pretraining, CSR can achieve significant gains of up to 52% on the more challenging inductive few-shot tasks where the entities are also unseen during (pre)training.

## 1 Introduction

Knowledge Graphs (KGs) are structured representations of human knowledge, where each edge represents a fact in the triplet form of (head entity, relation, tail entity) [10, 13, 17, 20]. Since KGs are typically highly incomplete yet widely used in downstream applications, predicting missing edges, *i.e.*, KG completion, is one of the most important machine learning tasks over these large heterogeneous data structures. Deep learning based methods have achieved great success on this task [19, 21, 26], but the more challenging few-shot setting [23] is much less explored: Given a background knowledge graph, an unseen relation, and a few support edges in triplet form, the task is to predict whether this unseen relation exists between a query entity and candidate answers based on the background knowledge graph. Such a setting captures the most difficult and important case during KG completion: predict rare relations (*i.e.* appearing only a few times in the existing KG) and incorporating new relations into the KG efficiently. It also tests the inductive reasoning skill of the model on deriving new knowledge data-efficiently, which is critical for AI in general.

---

[*]indicates equal contribution.

36th Conference on Neural Information Processing Systems (NeurIPS 2022).

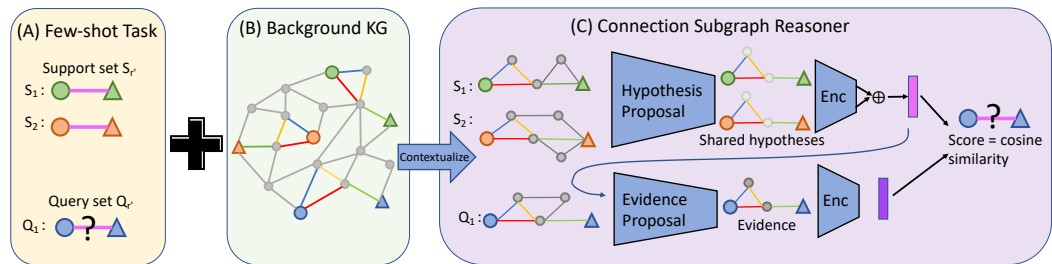

Figure 1: The few-shot KG completion problem includes (A) few-shot task that aims to learn a new relation (purple) and (B) background knowledge graph. Our CSR framework (C) first contexualizes all triplets in the background KG, then finds the shared hypothesis in the form of a connection subgraph using the Hypothesis Proposal module, and finally tests whether there is an evidence close enough to the hypothesis using Evidence Proposal module. In general all edges shown have different relation types, but here we only highlight ones in the connection subgraph with colors.

Existing approaches to few-shot KG completion [3, 18, 23, 24] typically adopt the meta-learning framework [8], where the model is trained over a meta-training set consisting of many few-shot tasks created from different relations in the background knowledge graph. GMatching [23], FSRL [24] and Att-LMetric [18] are metric-based meta-learning methods that try to learn a good metric where positive query pairs are closer to representation of edges in the support set than the negative ones. MetaR [3] is an optimization-based meta-learning method that use a meta-learner to improve the optimization of the task learner, such that the task learner can quickly learn with only few examples.

However, creating the meta-training set for some unknown few-shot tasks in test time is a very difficult *ad hoc* process in practice. On existing benchmarks [23], the training few-shot tasks and the target few-shot tasks are both randomly sampled from relations with least occurrences in the full knowledge graph, meaning the training and target relations are from the same distribution. But in reality, one has no information about the target few-shot tasks and the meta-training needs to be manually constructed out of the background knowledge graph. This is challenging since background knowledge graph often has a limited number of tasks due to the limited number of relations; creating too many meta-training tasks out of the background KG may remove a large number of edges from the KG, making it sparse and hard to learn over. Moreover, with a small meta-training set, the target few-shot tasks are very likely out of the curated meta-training set distribution, since the novel relation could be more complicated than known ones and the entities involved the target few-shot tasks can also be unseen. This then makes meta-learning based method suffer negative transfer due to distribution shift. Thus, having a method that can perform well on any novel few-shot tasks without relying on specifically designed meta-training set is crucial for real-world applications.

Here we propose a novel modeling framework *Connection Subgraph Reasoner (*CSR*)* that can make prediction on the target few-shot task directly without the need for meta-learning and creation of a curated set of training few-shot tasks. Our insight is that a triplet of the unseen relation of interest can be inferred through the existence of a *hypothesis* in the form of a *connection subgraph*, *i.e.* a subgraph in KG that connect the two entities of the triplet. Intuitively, the *connection subgraph* represents the logical pattern that implies the existence of the triplet. For the (chop, ⋈, kitchen) example, such a connection subgraph that implies ⋈ is a two hop path in KG: {(chop, `can be done with`, knife), (knife, `is located at`, kitchen)}. This insight allows us to cast the few-shot link prediction as an inductive reasoning problem. Following the eliminative induction method of inductive reasoning [9], our framework first recovers this hypothesis from the support triplets by finding the connection subgraph approximately shared among the support triplets, then tests whether this hypothesis is also a connection subgraph between the query entity and a candidate answer. We show the full pipeline along with an example of connection subgraph in Figure 1. To better adapt to specific KG, we design a novel encoder-decoder architecture based on graph neural networks (GNN) to implement the two stages and a corresponding self-supervised pretraining scheme to reconstruct diverse connection subgraphs.

We demonstrate that a training-free implementation of CSR via edge mask optimization can already discover the connection subgraph and reach link prediction performance competitive to many meta-learning methods over real-world knowledge graphs. With pretraining and optionally meta-learning over background KG uniformly, our method achieves high performance on both transductive and inductive few-shot test tasks that involve long tails relations, which are out of distribution to the training tasks; while existing methods using meta-learning suffers from distribution shift and cannot handle inductive tasks. Over real KGs including NELL [13], FB15K-237 [20], and ConceptNet [17], our method consistently exceeds or matches state-of-the-art methods in meta-training tasks free setting, and far exceeds the best existing methods by up to 52 % in the the more challenging inductive few-shot tasks where entities in the target few-shot tasks are also unseen. The implementation of CSR can be found in https://github.com/snap-stanford/csr.

## 2   Related Work

### 2.1   Few-shot Relational Learning via Meta-Learning

Meta-learning is a paradigm of learning across a set of meta-training tasks and then adapting to a new task during meta-testing [8]. To the best of our knowledge, all existing methods on few-shot KG completion follow the meta-learning paradigm to address the data scarcity in the target few-shot task [3, 18, 23, 24]. Therefore, these methods require the access to a meta-training set that contains many few-shot KG completion tasks for training. On the two existing benchmarks NELL-One and Wiki-One[23] the meta-training set is constructed by sampling from long tail relations, in the same way as the target few-shot tasks are constructed. However, such a meta-training set is not given in real world application and needs to be manually constructed out of the background knowledge graph $\mathcal{G}$ to mimic the actual few-shot task during test time. This curation is inherently challenging because that the background knowledge graph has a limited number of relations/tasks in $\mathcal{G}$, the distribution of the novel relations of interest is unknown, and the entites in the target few-shot tasks can be unseen in the background KG. In this paper, we develop a more general pretraining procedure to remove the dependency on manually created training tasks.

### 2.2   Few-shot Learning via Pretraining

It has been shown in natural language processing [2, 16] and computer vision [4, 5, 7] domains that large-scale self-supervised pretraining can significantly improve task-agnostic few-shot learning ability. One of the most successful pretraining objectives is predicting the next token or image patch given ones seen before it. However, how to design such powerful pretraining objectives for few-shot relational learning is still under-explored. In this work, we design a well motivated self-supervised pretraining objective, *i.e.* recovering diverse connection subgraphs that correspond to different inductive hypothesis. We show that such a pretraining scheme can significantly improve few-shot relational learning tasks on knowledge graphs.

## 3   Few-shot KG Completion

Few-shot KG completion is defined as follows [3, 23]: Denote the background KG that represents the known knowledge as $\mathcal{G} = (\mathcal{E}, \mathcal{R}, \mathcal{T})$, where $\mathcal{E}$ and $\mathcal{R}$ represents the set of entities and relations. $\mathcal{T} = \{(h, r, t)|h, t \in \mathcal{E}, r \in \mathcal{R}\}$ represents the facts as triplets. Given a new relation $r' \notin \mathcal{R}$ and a support set $S_{r'} = \{(h_k, r', t_k)|h_k \in \mathcal{E}\}_{k=1}^{K}$, we want to make predictions over a query set $Q_{r'} = \{(h_j, r', ?)|h_j \in \mathcal{E}\}_{j=1}^{J}$. This prediction on $(h_j, r', ?)$ is typically converted to scoring triplet $(h_j, r', e)$ for all candidate entities $e$ then ranking the scores. So we will proceed to consider $Q_{r'}$ as directly containing full triplets $(h_j, r', e)$ to score. We call this a $K$-shot KG completion task, typically the number of support is a small number ($K \leq 5$).

Note that existing works generally assume the entities in the few-shot tasks (support + query set) belong to the background KG. However, in real world cases, the goal of few-shot KG completion is to simulate learning of novel relations that may involve new entities not exist yet on the KG. Thus, in this paper we also consider a more challenging inductive setting where entities in the few-shot tasks do not belong to the entity set $\mathcal{E}$, but new triplets about these unseen entities can be added at test time.

# 4  Connection Subgraph Reasoner

In this section, we first discuss our main motivation from the inductive reasoning perspective and present the general framework based on it. Then we introduce both learning-free and learning-based implementations of this framework.

## 4.1  Inductive Reasoning

Inductive Reasoning refers to the reasoning process of synthesizing a general principle from past observations, and then using this general principle to make predictions about future events [9]. Few-shot link prediction task can be seen as an inductive reasoning task with background knowledge.

The key motivation of our work is eliminative induction, one of the principled methods used to reach inductive conclusions. Specifically, we consider the scientific hypothesis method: eliminating hypotheses inconsistent with observations. In the context of few-shot link prediction task, we explicitly try to find a hypothesis consistent with all examples in the support set, then test whether the the query is consistent with this hypothesis.

To illustrate a simple case of this, we use the $\bowtie$ example where the support triplets are (chop, $\bowtie$, kitchen), (read, $\bowtie$, library), and query triplet is (sleep, $\bowtie$, ?). From a background KG (*e.g.*ConceptNet), we can know a lot of knowledge in forms of triplets about these entities, such as (kitchen, is part of, a house) and (read, is done by, human) etc. We essentially want to find an induction hypothesis that explains how chop is related to kitchen in the same way that read is related to library. In other words, we want to find the shared connection pattern over the background KG that connects both two pairs of entities. In this case, we can observe that there is a simple shared 2 hop connection path that connects both pairs:

$$\{(\texttt{chop}, \texttt{can be done with}, \texttt{knife}), (\texttt{knife}, \texttt{is located at}, \texttt{kitchen})\} \tag{1}$$

$$\{(\texttt{read}, \texttt{can be done with}, \texttt{book}), (\texttt{book}, \texttt{is located at}, \texttt{library})\} \tag{2}$$

The abstracted inductive hypothesis consistent with both examples in the support set is then

$$\exists Z, (h_c, \texttt{can be done with}, Z) \wedge (Z, \texttt{is located at}, t_c) \implies (h_c, \bowtie, t_c). \tag{3}$$

This hypothesis can then be used to deduce that (sleep, $\bowtie$, bedroom) has a high score, since we know $\{(\texttt{sleep}, \texttt{can be done with}, \texttt{bed}), (\texttt{bed}, \texttt{is located at}, \texttt{bedroom})\}$ from the background KG.

More generally, the shared connection pattern can be graph structured instead of a two-hop path, which then form a connection subgraph between the two end entities instead of a connection path (Figure 1). Here we define the connection subgraph: Let $\mathcal{G}' = (\mathcal{E}', \mathcal{R}', \mathcal{T}')$ be any subgraph of the background KG $\mathcal{G}$, (*i.e.*, $\mathcal{E}' \subseteq \mathcal{E}$, $\mathcal{R}' \subseteq \mathcal{R}$ and $\mathcal{T}' \subseteq \mathcal{T}$) that satisfies the following requirement for a given pair of nodes $(h_c, t_c)$ on the KG. (1) $h_c \in \mathcal{E}'$ and $t_c \in \mathcal{E}'$; (2) there is no disconnected component. We define the connection subgraph $\mathcal{G}_C$ of $(h_c, t_c)$ to be any such $\mathcal{G}'$ where we further ignore the node identity. The key insight is that we should only consider the relation structure patterns and abstract away the node identity in order to construct a hypothesis.

Then a hypothesis like Eq. 3 can be represented as a connection subgraph $\mathcal{G}_C$ by interpreting each clause as an edge. And such a hypothesis is *consistent* with a support/query triplet $(h, r, t)$ if $\mathcal{G}_C$ is a connection subgraph of $h, t$. In terms of the $\bowtie$ example, the triplet (sleep, $\bowtie$, bedroom) is consistent with the hypothesis Eq. 3 because the connection subgraph form of the hypothesis $(h_c, \texttt{can be done with}, Z) \wedge (Z, \texttt{is located at}, t_c)$ is a connection subgraph between sleep and bedroom, with $h_c, Z, t_c$ corresponding to sleep, bed and bedroom respectively. Given a pair of node, we further call the KG subgraph with node identity an *evidence* that witnesses why the hypothesis is consistent with a connection subgraph of $h, t$;. Note the key difference between hypothesis and evidence is that, different pair of nodes may share the same hypothesis but each may have different evidence to support the consistency with such a hypothesis. As an example, for both (sleep, $\bowtie$, bedroom) and (chop, $\bowtie$, kitchen), the hypothesis is the same $(h_c, \texttt{can be done with}, Z) \wedge (Z, \texttt{is located at}, t_c)$. Yet the evidences are respectively $\{(\texttt{sleep}, \texttt{can be done with}, \texttt{bed}), (\texttt{bed}, \texttt{is located at}, \texttt{bedroom})\}$ and $\{(\texttt{chop}, \texttt{can be done with}, \texttt{knife}), (\texttt{knife}, \texttt{is located at}, \texttt{kitchen})\}$

Note that this hypothesis formulation can be seen as a generalization of the typical path structured logic rules considered by the multi-hop reasoning [11, 12, 22] and rule induction [6, 15] literature.

However, even with our graph structured hypothesis so far, such a strict way of testing the consistency with a hypothesis by checking of whether $\mathcal{G}_C$ is a connection subgraph of $h, t$ exactly would be time-consuming and brittle due to the fuzziness of target relations and the KG incompleteness. Below we develop our framework that relaxes the hypothesis representation and consistency testing criteria.

## 4.2 General Framework

Based on the above motivation, we design the following general framework Connection Subgraph Reasoner (CSR) that includes 2 main modules: hypothesis proposal module $M_p$ and evidence proposal module $M_e$. CSR then has the following 3 components:

**(1) Triplet Contextualization.** We first contextualize each triplet $(h, r', t)$ in support set $S_{r'}$ and query set $Q_{r'}$ by retrieving its contextualized graph $G(h, t) \subset \mathcal{G}$ such that it contains $h, t$ and captures most of the immediately relevant information about the pair. There are several prior works on retrieving smaller subgraph around a triplet in KG for link prediction [19, 25]. We choose to use the enclosing subgraph proposed by Grail [19], which is the subgraph induced by all nodes that are in the $k$ hops neigborhood of both $h, t$. We generally use $k = 1, 2$ depends on the density of KG. We also supplement with random sampling of the neighbors of $h, t$ in case the enclosing subgraph itself is too small. We call the contextualized graphs of support triplets *support graphs*, and the contextualized graphs of query triplets *query graphs*.

**(2) Hypothesis Proposal.** After contextualization, we would like to find the hypothesis consistent with all support graphs. We use a hypothesis proposal module $M_p$ to generate a hypothesis from each support graph such that they are most similar to each other. Each hypothesis can be represented as a soft edge mask $m : [0, 1]^E$ over edges in the corresponding support graph. To aggregates these hypotheses to produce the embedding of one final hypothesis $b$, we take the mean of their GNN embedding produced by a graph encoder $M_{enc}$.

$$\{m_i\}_{i=1}^K = M_p(\{G(h_i, t_i) | (h_i, r', t_i) \in S_{r'}\})$$
$$b = \frac{1}{K} \sum_{i=1}^K M_{enc}(G(h_i, t_i), m_i) \tag{4}$$

$M_p$ should compare all support graphs and output masks that represent the largest common connection subgraph, *i.e.*

$$\arg\max(\sum_{i=1}^K \sum_E m_i),$$
$$s.t. \ \forall i, j \in 1...K, \ s(M_{enc}(G(h_i, t_i), m_i), \ M_{enc}(G(h_j, t_j), m_j)) > 1 - \epsilon \tag{5}$$

**(3) Hypothesis Testing.** Finally, we want to test whether each query graph is consistent with the proposed hypothesis. We uses evidence proposal module $M_e$ to take in $b$ and a query graph to output the closest evidence to the hypothesis represented by $b$. The score of the query is then the cosine similarity between $b$ and the embedding of the evidence.

$$m_q = M_e(b, G(h_q, t_q))$$
$$score = s(b, M_{enc}(G(h_q, t_q), m_q)) \tag{6}$$

$M_e$ is intended to output $m_q$ such that

$$s(M_{enc}(G(h_q, t_q), m_q), b) > 1 - \epsilon \tag{7}$$

Overall, we model hypothesis softly as the aggregation of approximately shared hypothesis connection subgraphs between support graphs as found by $M_p$, then test its probabilistic consistency with query graph by finding how close an evidence for this hypothesis can be found in the query graph by $M_e$.

For the $M_{enc}$, we use the alternative message passing in PathCon [21] so that that it does not use entity embedding as input can measures graph isomorphism :

$$a_v^i = \frac{1}{1 + \sum_{e \in \mathcal{N}(v)} m_e} \sum_{e \in \mathcal{N}(v)} s_e^i \cdot m_e \tag{8}$$

$$s_v^i = a_v^i || \mathbb{1}(v = h) || \mathbb{1}(v = t) \tag{9}$$

$$s_e^{i+1} = \sigma(([s_v, s_u, s_e]) \cdot W^i + b^i), u, v \in \mathcal{N}(e) \tag{10}$$

$$M_{enc}(G, m) = max\_pool(a_v^L) || a_h^L || a_t^L, \tag{11}$$

where $L$ is number of layers and $s_e^1$ is random or pretrained relation embedding. Note that we concatenate the head and tail representation in the final graph representation so that the hypothesis and evidence subgraph are compared with heads and tails matched already.

For $M_p$ and $M_e$, we introduce two specific implementations that satisfies the requirements Eq. 5 and 7 in the following sections.

### 4.3 CSR-OPT: Learning-free Implementation

We implement $M_p$ and $M_e$ as a learning free optimization processes, where the masks are optimized toward Eq. 5 and 7 directly.

For $M_p$, we formulate it as a constraint optimization problem:

$$M_p(\{G(h_i, t_i) | (h_i, r', t_i) \in S_{r'}\}) = \arg\max \sum (\{m_i\}_{i=1}^K) - \lambda * H(m_q) \tag{12}$$

$$s.t. \sum s(M_{enc}(G(h_i, t_i), m_i), M_{enc}(G(h_j, t_j), m_j)) > 1 - \epsilon',$$

$$\texttt{connectivity}(m_i) > 1 - \epsilon'$$

Here we add a constraint on the connectivity that measures whether the nodes in the subgraph represented by $m_i$ can be reachable from head or tail within 2 hops in the subgraph. Let $A$ be a soft adjacency matrix of the edge mask $m_i \in [0,1]^{|E|}$. We compute whether two nodes $i, j$ can reach each other within 2 hops as $R_{i,j} = min((I + A + A^2)[i,j], 1)$, then

$$\texttt{connectivity}(m_i) = \frac{1}{|E|} \sum_{e=(n,n') \in E} m_{ie} * \min(R_{n,h_i} + R_{n,t_i} + R_{n',h_i} + R_{n',h_i}, 1) \tag{13}$$

We also add the entropy regularization terms $H(\cdot)$ to force the mask to represent a valid subgraph. Similarly, for $M_e$:

$$T(b, G(h_q, t_q)) = \arg\max s(b, M_{enc}(G(h_q, t_q), m_q)) - \lambda * H(m_q) \tag{14}$$

We optimize each objective independently using gradient descent, but do not train them together end to end. The $M_{enc}$ remains random initialized, but we found that this "random" GNN is already powerful enough to distinguish non-isomorphic graphs as needed. Although this implementation is straight forward and learning free, it is slow and difficult to incorporate additional training. Nevertheless, it serves as a good way for directly verifying our hypothesis on the framework design.

### 4.4 CSR-GNN: Learning-based Implementation and Pretraining Scheme

We also design a fully GNN-based encoder-decoder approach to implement $M_p$ and $M_e$. We use two models as building blocks: encoder $f_{\text{ENC}}(G_1, m_1) : \mathcal{G} \times \mathbb{R}^{|\mathcal{E}|} \to \mathbb{R}^d$ that encodes graph $G_1$ weighted by $m_1$ to an embedding $b$, and decoder $f_{\text{DEC}}(G_2, b) : \mathcal{G} \times \mathbb{R}^d \to \mathbb{R}^{|\mathcal{E}|}$ that decodes out the weight $m_2$ such that graph $G_2$ weighted by $m_2$ corresponds to the input embedding $b$. Intuitively, $f_{\text{DEC}}(G_2, f_{\text{ENC}}(G_1, m_1)))$ compares $G_1, G_2$ and finds the subgraph of $G_2$ that is closest to the subgraph of $G_1$ induced by $m_1$.

We then implement $M_p$ as an iterative process of comparing between all pairs of support graphs using this encoder-decoder, as shown in Algorithm 1. We start with a full edge mask $m_i = \mathbf{1}$ for each graph $G_i$. During each iteration, each graph $G_j$ obtains edge mask $m_{jk}$ as a result of comparing against $G_k$ weighted by $m_k$. $G_j$ then takes the shared parts between $m_{jk}$ by taking an element wise minimum to obtain $m_j$.

---

**Algorithm 1** Hypothesis Proposal Module $M_p$ of CSR-GNN

---

**Require:** $n$ support graphs $G_1, \ldots, G_n$
 1: Initialize the masks of all support graphs to be all ones: $m_i = \mathbf{1}, \forall i \in 1, \ldots, n$
 2: **for** $iter \leftarrow 1, \ldots$ **do**
 3:     **for** $j \leftarrow 1, \ldots, n$ **do**
 4:        **for** $k \leftarrow 1, \ldots, n$ **do**
 5:           $m_{jk} = f_{\text{DEC}}(G_j, f_{\text{ENC}}(G_k, m_k))$
 6:        **end for**
 7:        $m_j = \min_k m_{jk}$
 8:     **end for**
 9: **end for**
10:
11: **return** $[m_1, \ldots, m_n]$

---

**Algorithm 2** CSR-GNN Full Architecture

---

**Require:** $n$ support graphs $G_1, \ldots, G_n$, query graph $G_q$
 1: $m_1, \ldots, m_n = M_p(G_1, \ldots, G_n)$
 2: Obtain subgraph embedding using the encoder: $g_i = f_{\text{ENC}}(G_i, m_i)$
 3: Average subgraph embedding of the supporting graphs $b = \frac{1}{n} \sum_i g_i$
 4: Decode masks using $g$ from the query graph $G_q$: $m_q = f_{\text{DEC}}(G_q, b)$
 5: Obtain subgraph embedding for the query graph: $g_q = f_{\text{ENC}}(G_q, m_q)$
 6:
 7: **return** cosine_similarity$(g_q, b)$

---

We use $f_{\text{DEC}}(\cdot)$ directly as $M_e$. Combining these two steps, the full architecture is shown in 2, where the encoder $f_{\text{ENC}}(\cdot)$ is shared with $M_{enc}$. We use the same PathCon atchitecture for $f_{\text{DEC}}(\cdot)$ as for $M_{enc}$, except that we concatenate $b$ to the input edge embeddings before the first layer.

**Encoder-Decoder Pretraining** To train this encoder-decoder architecture, we simply use reconstruction: given a graph $G$ and a randomly sampled mask $m \in \mathbb{R}^{|\mathcal{E}|}$, we should obtain back the same mask $m$ after applying both encoder and decoder (Figure 2)

$$Loss_{recon} = \ell_{\text{CE}}(m, f_{\text{DEC}}(G, f_{\text{ENC}}(G, m))) \tag{15}$$

To jointly train together with $M_{enc}$, we also add contrastive loss

$$g = M_{enc}(G, m) \tag{16}$$

$$g_{pos} = M_{enc}(G, f_{\text{DEC}}(G, f_{\text{ENC}}(G, m))) \tag{17}$$

$$g_{neg} = M_{enc}(G', f_{\text{DEC}}(G', f_{\text{ENC}}(G, m))) \tag{18}$$

$$Loss_{contrast} = \max(s(g_{pos}, g) - s(g_{neg}, g) + \gamma, 0) \tag{19}$$

During pretraining, we sample graph $G$ and $G'$ by sampling random triplets per relation and contextualize them. $G, G'$ should correspond to different relations. Then we sample $m$ that represents mutiple random paths connecting to either head or tail in $G$.

## 5 Experiments

We evaluate our method CSR-OPT and CSR-GNN on few-shot KG completion tasks over three real world KGs – NELL, FB15K-237 and ConceptNet. We take NELL directly from NELL-One [23] but

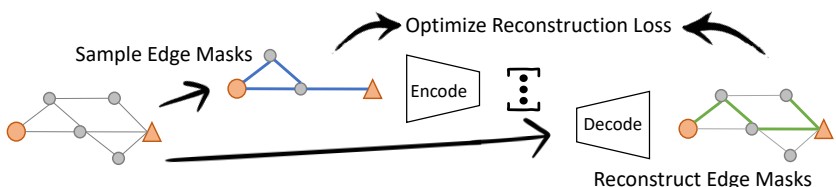

Figure 2: Pretraining of Connection Subgraph Reconstruction.

Table 1: Statistics of the benchmark datasets. BG refers to the background KG available during training time.

| | NELL | | | | FB15K-237 | | | | ConceptNet | | | |
| | #rels | #entities | #edges | #tasks | #rels | #entities | #edges | #tasks | #rels | #entities | #edges | #tasks |
|---|---|---|---|---|---|---|---|---|---|---|---|---|
| Trans-BG | 291 | 68544 | 181109 | 11 | 200 | 14543 | 268039 | 30 | 14 | 790703 | 2541996 | 2 |
| Ind-BG | 291 | 44005 | 82318 | - | 200 | 11290 | 112477 | - | 14 | 619163 | 1191782 | - |
| Ind-Test | 291 | 24539 | 98791 | 11 | 200 | 3253 | 155562 | 10 | 14 | 171540 | 1350214 | 2 |

add different settings; we select the fewest appearing relations as target few-shot tasks in FB15K-237 and ConceptNet following [12, 23]. We summarize the statistics of all three datasets in Table 1. We also created one synthetic dataset for explanatory purpose. Across all four KGs, (1) we first evaluate our method against the previous methods on real KGs in terms of few-shot learning performance without the pre-designed training few-shot tasks set, on both transductive and inductive settings; (2) we perform extensive ablation studies on each component of our method and show that every component are indispensable in our framework; (3) we show that with a manually curated meta-train set, meta-learning based method suffers from negative transfer when facing a different target test set; (4) on the synthetic dataset, we further show that our method is able to recover complex hypothesis and evidence connection subgraphs during the prediction. We consider state-of-the-art few-shot KG completion baselines MetaR and FSRL in terms of the standard ranking metrics MRR and Hits@$h$, where we sample 50 negative tail candidates for each query triplet and rank them together with the positive tail entity. Hits@$h$ measures the percentage of times that the positive tail is ranked higher than $h$ among the negative tail candidates. We use $h = 1, 5, 10$. For simplicity, we only consider the number of few shot example $K = 3$, even though all methods here can generalize to arbitrary $K$. See appendix A for more details on full experiment setups. Since the methods have low variance in general, bellow we omit the standard deviations in the table and provide them instead in appendix A.

## 5.1 Few-shot Learning without Curated Training Tasks

We first evaluate our method CSR-OPT and CSR-GNN along with baselines on the three real KGs without curated training tasks. This means that on NELL we do not use the meta-train split originally provided in NELL-One. To adapt MetaR and FSRL to this setting, we use pretrained entity and relation embeddings as in the original papers, but meta-train them on randomly sampled tasks from the background KG instead. Since our method is designed to not include entity embeddings, we add it in by concatenating the head and tail entity embedding to the representation produced by $M_{enc}$ in CSR-GNN. Similarly, we also perform end-to-end finetuning on the same set of randomly sampled tasks in addition to our pretraining objectives. See ablations for these modifications in Section 5.2.

### 5.1.1 Transductive Setting

As shown in Table 2, we demonstrate that our learning-free method CSR-OPT can already give competitive performance without any training over the real dataset. With CSR-GNN, we can pretrain over these real KG to achieve higher performance exceeding/competitive to meta learning results: CSR-GNN gives 17.8% improvement of MRR over the second best method on NELL, 5% improvement on ConceptNet and comes close second on FB15k-237. On FB15K-237, the graph is much denser than NELL and ConceptNet so that the pretrained entity embeddings could already capture most relational structures when predicting the query triples. However, this is only limited to the transductive setting, where all entities are seen during pretraining.

### 5.1.2 Inductive Setting

We evaluate the same set of methods in inductive version of these three datasets, where all entities involved in the testing few-shot task and their one hop neigbors are unseen in the background knowledge graph. In this setting, we do not use entity embedding for our methods, and pick the best performance for baselines between using and not using entity embedding. As shown in table 5, our methods only drops slightly in performance than in the transductive setting comparing to baselines, resulting in significantly larger gains of up to 52% in this more challenging setting. This is because our architecture is designed entirely based on topological rule and does not rely on entity embeddings, while the performance of baselines in the transductive setting rely heavily on entity embedding. To

Table 2: Performance comparison on transductive few-shot tasks without curated training tasks

|  |  | MRR | Hits@1 | Hits@5 | Hits@10 |
|---|---|---|---|---|---|
| NELL | MetaR | 0.471 | 0.322 | 0.647 | 0.763 |
|  | FSRL | 0.490 | 0.327 | 0.695 | 0.853 |
|  | CSR-OPT | 0.463 | 0.321 | 0.629 | 0.760 |
|  | CSR-GNN | **0.577** | **0.442** | **0.746** | **0.858** |
| FB15K-237 | MetaR | **0.805** | **0.740** | **0.881** | **0.937** |
|  | FSRL | 0.684 | 0.573 | 0.817 | 0.912 |
|  | CSR-OPT | 0.619 | 0.512 | 0.747 | 0.824 |
|  | CSR-GNN | 0.781 | 0.718 | 0.851 | 0.907 |
| ConceptNet | MetaR | 0.318 | 0.226 | 0.390 | 0.496 |
|  | FSRL | 0.577 | 0.469 | 0.695 | 0.753 |
|  | CSR-OPT | 0.559 | 0.450 | 0.692 | 0.736 |
|  | CSR-GNN | **0.606** | **0.496** | **0.735** | **0.777** |

Table 3: Inference time on NELL (inductive setting).

| NELL (inductive) | MetaR | FSRL | CSR-GNN |
|---|---|---|---|
| Inference Time (s) | 5.52 | 14.24 | 17.50 |
| MRR | 0.355 | 0.180 | 0.511 |

Table 4: Inference time on NELL (transductive setting).

| NELL (transductive) | MetaR | FSRL | CSR-GNN |
|---|---|---|---|
| Inference Time (s) | 5.50 | 14.49 | 18.33 |
| MRR | 0.471 | 0.490 | 0.577 |

further demonstrate this, we show in Appendix B that our methods achieve similar large gain of up to 52% under transductive setting when all entity embedding are randomized during testing, equivalent to an extreme inductive setting.

Table 5: Performance comparison on inductive few-shot tasks without curated training tasks

|  |  | MRR | Hits@1 | Hits@5 | Hits@10 |
|---|---|---|---|---|---|
| NELL | MetaR | 0.353 | 0.191 | 0.517 | 0.820 |
|  | FSRL | 0.180 | 0.090 | 0.242 | 0.360 |
|  | CSR-OPT | 0.425 | 0.303 | 0.534 | 0.657 |
|  | CSR-GNN | **0.511** | **0.348** | **0.725** | **0.837** |
| FB15K-237 | MetaR | 0.315 | 0.143 | 0.506 | 0.896 |
|  | FSRL | 0.453 | 0.299 | 0.571 | **0.922** |
|  | CSR-OPT | 0.554 | 0.429 | 0.727 | 0.844 |
|  | CSR-GNN | **0.624** | **0.479** | **0.833** | 0.894 |
| ConceptNet | MetaR | 0.154 | 0.041 | 0.260 | 0.452 |
|  | FSRL | 0.402 | 0.233 | 0.603 | 0.740 |
|  | CSR-OPT | 0.547 | 0.425 | 0.726 | 0.740 |
|  | CSR-GNN | **0.611** | **0.496** | **0.729** | **0.786** |

### 5.1.3 Inference Time

We have also measured the inference time of our method and baselines on NELL. As shown in Table 3 and 4, we find that our method has comparable inference runtime compared with state-of-the-art baselines FSRL but it's slower than MetaR. The reason is that for each query triplet, our model needs to use the evidence proposal module to decode the edge masks and obtain the embeddings for the connection subgraphs. Compared with MetaR, which directly uses shallow KG embeddings to score each query triplet in TransE style, CSR achieves much better empirical performance and also allows inductive few-shot KG link prediction.

Table 6: Ablative results on NELL.

| Entity Embedding | Finetuning | Hypothesis Proposal | Evidence Proposal | MRR |
|:---:|:---:|:---:|:---:|:---:|
| ✔ | ✔ | ✔ | ✔ | 0.577 |
| ✗ | ✔ | ✔ | ✔ | 0.511 |
| ✗ | ✗ | ✔ | ✔ | 0.466 |
| ✗ | ✗ | ✗ | ✔ | 0.441 |
| ✗ | ✗ | ✔ | ✗ | 0.436 |
| ✗ | ✗ | ✗ | ✗ | 0.420 |

## 5.2 Ablation Study

Here we conduct ablation study on each component of our proposed model CSR-GNN on the NELL dataset. Specifically we consider the four components of our models. It includes (1) whether we use the pretrained KG entity embeddings or not; (2) whether we perform additional finetuning of our model; (3) whether we perform the hypothesis proposal as shown in Algorithm 1, an alternative is to directly assign full one masks for $m_1, \ldots, m_n$ without detecting the common subgraph; (4) whether we perform the evidence proposal for the query triplets, if not, we will propose a full one mask for the query graph. As shown in Table 6, each component in our model is indispensable. Due to our model architecture design, without entity embeddings, MRR decreases from 0.577 to 0.511, however it's still much higher than state-of-the-art baselines with pretrained entity embeddings (0.471 for MetaR). Next we show that the two proposed stages Hypothesis Proposal and Evidence Proposal play a key role. Without either of them, it means that our model can no longer accurately perform eliminative induction over the support or the query triples, which may significantly deteriorates the performance.

## 5.3 Robustness to Distribution Shift

To demonstrate the distribution shift problem when constructing meta-training set, we construct a new set of test few-shot learning tasks and compare the meta-learning based methods performance on this new test set against performance on the original test set when using the same meta-training set. Specifically, on the existing NELL dataset, the meta-training and test tasks/relations are sampled from the same long-tail distribution. Here we sample the new set of test few-shot learning tasks by randomly sampling the 10 test relations from the whole background KG, so that there exists a gap between the training tasks and the test tasks on the new set. Comparing ours with the state-of-the-art FSRL, we achieve comparable MRR on both test the original and our new challenging tasks (0.590 vs 0.540) while FSRL suffers greatly from the distribution gap (0.578 vs 0.459). This demonstrates the robustness of our method in handling different distribution of test tasks.

## 5.4 Synthetic Dataset

For explanatory purpose, we construct synthetic datasets that strictly follow our assumptions so that we have the ground truth of the hypothesis connection subgraphs. In each task, all support graphs contain a shared connection graph, and the query is only True if the query graph also contains the same connection graph. We show that CSR-OPT can recover hypothesis and evidence connection subgraphs with high IOU of 0.843 and 0.992 when the hypothesis is in the form of a 4 clique. With direct supervision of ground truth hypothesis during training, CSR-GNN can also achieve high IOU of 0.809 and 0.981. See full details in Appendix C.

## 6 Conclusion

In this paper we proposed CSR, a general framework for few-shot relational reasoning over knowledge graphs using self-supervised pretraining. Based on eliminative induction, we model hypothesis as the shared connection subgraph between the support triplets and predict the query triplets by checking evidence of the connection subgraph in the query graph. Our method achieves state-of-the-art performance across multiple datasets on few-shot link prediction without curated training tasks.

## Acknowledgments and Disclosure of Funding

We thank Yuhuai Wu for discussions and for providing feedback on our manuscript. We also gratefully acknowledge the support of DARPA under Nos. HR00112190039 (TAMI), N660011924033 (MCS); ARO under Nos. W911NF-16-1-0342 (MURI), W911NF-16-1-0171 (DURIP); NSF under Nos. OAC-1835598 (CINES), OAC-1934578 (HDR), CCF-1918940 (Expeditions), NIH under No. 3U54HG010426-04S1 (HuBMAP), Stanford Data Science Initiative, Wu Tsai Neurosciences Institute, Amazon, Docomo, GSK, Hitachi, Intel, JPMorgan Chase, Juniper Networks, KDDI, NEC, and Toshiba. Hongyu Ren is supported by Masason Foundation PhD fellowship, Apple PhD fellowship and Baidu PhD scholarship. Qian Huang is supported by Open Philanthropy AI fellowship.

The content is solely the responsibility of the authors and does not necessarily represent the official views of the funding entities.

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
