# Appendix

## A Experiment Setup

In this section, we present full experiment setups and results on the three real KGs in detail.

### A.1 Data Splits

We use NELL, FB15K-237 and ConceptNet for both transductive and inductive setting. For the transductive setting, we use the meta-eval and meta-test splits of NELL-One for the eval and test few-shot tasks on NELL, and we do not use the meta-train split. For FB15K-237 and ConceptNet, we select the fewest 7:30 and 1:2 appearing relations as eval:test few-shot tasks respectively, following the previous papers [12, 23]. The background KGs are generated by removing all triplets involving eval and test relations.

For the inductive setting, we mostly use the same set of eval and test relations, except for FB15K-237 where we randomly selected 10 relations out of the 30 transductive test relations as the inductive test relations. For each test task, we also subsample the number of query triplets to 10%. Then we consider all entities and their one hop neighbors appeared in test tasks the inductive entities unseen during training time. So all inductive entities and all triplets involving them become the Ind-Test in 1 and are removed from the training time background KG – Ind-BG 1. During training time, only the Ind-BG is available as the background KG; during the test time, the Ind-Test is combined with the Ind-BG to form the test time background KG. Note that the subsamplings of query triplets and tasks are intended to make sure that the remaining training time background KG does not become too small and still contains all of the relations not in eval and test relations (which includes all the relations in Ind-Test).

We will release our data processing scripts and preprocessed datasets publicly for reproducibility.

### A.2 Model Architectures and Hyperparameters

#### A.2.1 Baselines

For MetaR and FSRL, we use their publicly available code directly and use the architectures and hyperparameters on NELL for other datasets and settings.

#### A.2.2 CSR

We include our code for CSR in the supplementary material and will release them publicly for reproducibility. The anonymous code and data can be found in the anonymous link `https://drive.google.com/file/d/18otchItFQurlHzQI2xQILudTzRCHlcDa/view?usp=sharing`.

**Triplet Contextualization** For the triplet contextualization step of CSR-GNN and CSR-OPT, we use $k = 2$ hops enclosing subgraph for NELL, $k = 1$ for FB15K-237 and ConceptNet. For all datasets, we supplement maximum of 50 randomly selected one hop neighbors of head and tail entities.

**Architectures and Hyperparameters** For CSR-GNN, both $f_{\text{ENC}}(\cdot)$ and $f_{\text{DEC}}(\cdot)$ use 3 layers of alternative message passing and hidden dimension of 128. For $f_{\text{DEC}}(\cdot)$, instead of having a global pooling at the end like in $f_{\text{ENC}}(\cdot)$, we apply a 2 layers MLP binary classifier (hidden dimension 64) to the edge representation produced by last layer message passing to generate the edge mask value on each edge. We use AdamW optimizer, 1e-5 learning rate, 5000 epochs, and linear decay learning rate schedule. The three loss terms are generally combined as $\lambda_1 * Loss_{recon} + \lambda_2 * Loss_{contrast} + Loss_{finetune}$. On NELL, $\lambda_1 = 0.7, \lambda_2 = 0.1$; On FB15K-237, $\lambda_1 = 0.1, \lambda_2 = 1$; On ConceptNet, $\lambda_1 = 2, \lambda_2 = 0.5$. We manually select these hyperparameters based on transductive eval set performance.

For CSR-OPT, the architecture for $M_{enc}$ is the same as $f_{\text{ENC}}(\cdot)$. We use gradient descent with AdamW for both optimizations. For the constraint optimization implementing $M_p$, we use the Basic Differential Multiplier Method [14]. All the hyperparameter are automatically searched using Optuna

[1] against the eval set performance in terms of the overall AUC-ROC when each query triplet is compared again only one negative candidate. Since $M_{enc}$ is not trained, CSR-OPT only uses relation embedding and does not use entity embedding under any settings.

**Entity and Relation Embedding** All methods use 100 dimensional relation and entities embedding when applicable. For transductive NELL dataset, we use the originally released embedding by [23]. For all other settings, we train 100 dim embedding over the appropriate (training) background KG and use them for all methods. Based on eval set performance, we use TransE embedding for MetaR and ComplEx embedding for FSRL, which makes sense since MetaR is designed with TransE distance and FSRL also reports best performance with ComplEx in the original paper. For CSR, we use TransE for NELL and FB15K-237 and ComplEx for ConceptNet.

## A.3   Full Results

We report full results with standard deviations for Table 2, 5 and 6. Note that in the inductive setting, the unseen entities are given a random embedding, following the embedding initialization distribution.

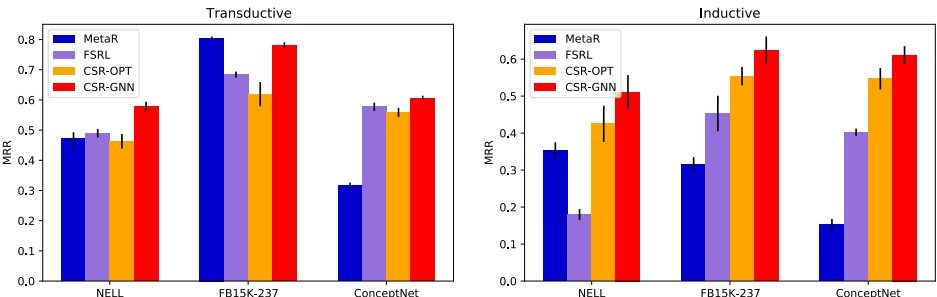

Figure 3: Full results with standard deviations for transductive (left) and inductive (right) settings, corresponding to Table 2 and 5.

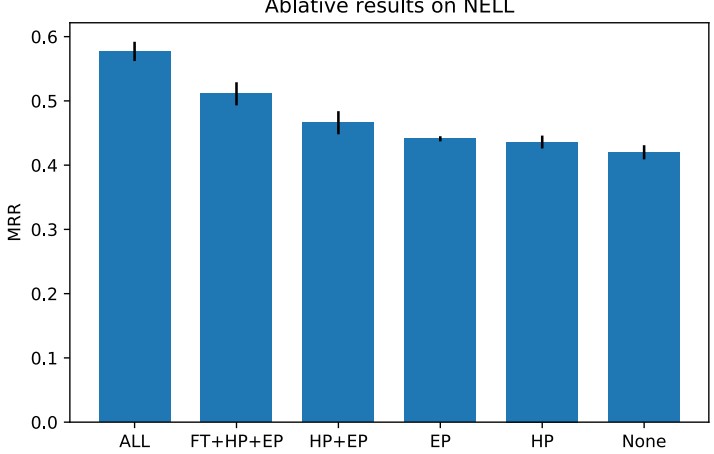

Figure 4: Full results with standard deviation of Table 6. Each category corresponds to one row in table 6: ALL corresponds to the first row with all four components; None corresponds to the last row without any of the four components. FT = Finetuning; HP= Hypothesis Proposal; EP = Evidence Proposal.

# B  Transductive Setting with no Entity Embedding During Testing

In figure 5, we randomize entity embedding during testing with transductive dataset, which can be seen as an extreme inductive setting where all entities are new during testing.

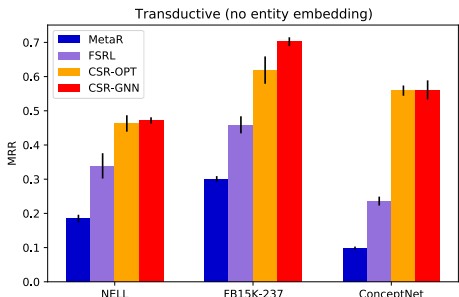

Figure 5: Performance comparison on transductive few-shot tasks without curated training tasks and with randomized entity embedding during testing.

# C  Synthetic Dataset

## C.1  Synthetic Dataset Construction

We construct synthetic datasets so that in each few-shot task the support graphs strictly contain a shared connection subgraph, and the query triplet is only correct if the query graph also contains the same connection subgraph. Specifically, we first sample a hypothesis in the form of a four clique with the link between head and tail missing. Here each edge has a relation randomly selected from 50 relations. We then augment this hypothesis graph to 100 different support and query graphs by first adding more nodes edges and randomly then pruning away the graph until each node is reachable to both head and tail node in 2 hops. Each few-shot task is then sampled from these generated support and query graphs and corresponds to a different hypothesis connection subgraph.

## C.2  Connection Subgraph Detection

Since all support graphs and (positive) query graphs in each task in the synthetic dataset have a shared connection graph, here we evaluate our method to see whether we are able to detect the common connection graph for both the support and query graphs.

For evaluating the detection of common hypothesis connection subgraph among support graphs, we use the training tasks and supervise our model with the ground truth common connection subgraph of the given support graphs in the training task. Then we evaluate whether our model is able to discover the common connection subgraph for the support graphs in the test tasks. In detail, given several support graphs in a task and the ground truth binary edge mask (applying which returns the connection subgraph), we calculate the intersection over union (IOU) of the ground truth edge mask and the predicted edge mask produced by our model. For hypothesis proposal, we are able to achieve an IOU of 0.843 and 0.809 for CSR-OPT and CSR-GNN respectively.

For evaluating the detection of evidence connection subgraph in query graphs, the setting is that the model is given the query graph and the connection subgraph embedding (achieved from the support graphs), we evaluate whether the model is able to predict the evidence connection subgraph in the query graph. Similar to the hypothesis proposal, we also calculate the IOU as the metric. We are able to achieve a 0.992 and 0.981 IOU for CSR-OPT and CSR-GNN respectively.

We demonstrate that on synthetic dataset, our method is able to automatically detect the common connection graphs on new few-shot tasks. This shows that our method has significantly better inductive bias and interpretability than prior methods as we are able to detect complex graph structured rule, which further leads to improvement in the downstream few-shot link prediction performance.

## D    Interpretable edge-level masks

We further explore qualitative examples and attach some visualizations of the connection subgraph discovered by our model in the NELL test set. Each figure visualizes the evidence connection subgraph discovered for the novel relation in the title. We include only relation names but not entity names to emphasize the topological similarity and reduce cluttering.

In these figures, we can see that the model discovered meaningful evidence connection subgraphs to support its prediction, *e.g.*,

> concept:agriculturalproductgrowninlandscapefeatures,
>
> concept:geopoliticallocationcontainscountry =>
>
> concept:agriculturalproductcamefromcountry.

The model also selects some one-hop neighbors of head/tail that are not reachable to the tail/head but provide information about the type of the head/tail entity: *e.g.* concept:countrycities helps to determine that the head entity is a country. Moreover, different subgraphs with similar semantics are identified when the exact same subgraph is not available.

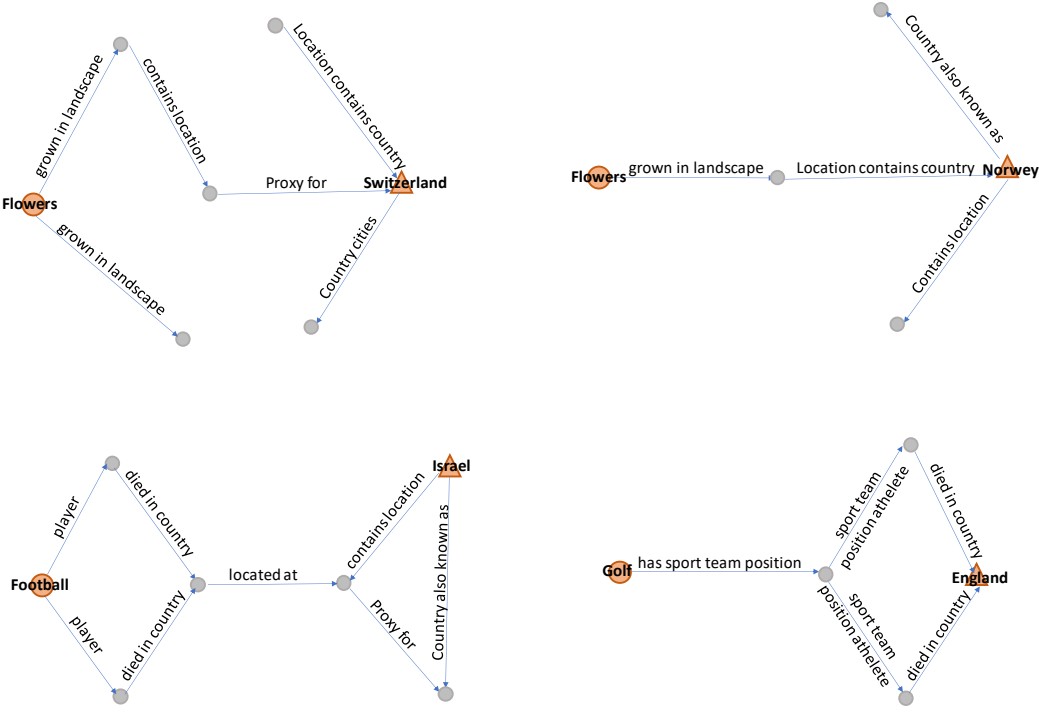

Figure 6: Learned edge masks by CSR. Top two are concept:agriculturalproductcamefromcountry; bottom two are concept:sportschoolincountry

## E    Computation

We use NVIDIA 2080 Ti RTX 11GB GPUs in our internal cluster for all of our model training and testing. Training for both CSR-GNN and baseline methods take around 2 hours on a single GPU.

## F    Limitations

Here we discuss the limitation of our method. Our method is extremely flexible in both the transductive and inductive few-shot link prediction on knowledge graphs. However, when the entity embedding

can already capture most of the relevant topological information, our method of explicitly modeling and comparing connection subgraphs could bring less improvement. This happens most naturally in transductive setting with a dense background KG, where each pair of query entities are both seen and already relatively close. Our method also relies on the triplet contextualization step to first provide a reasonable super set of possible hypothesis to consider. In the main paper, we uses enclosing subgraph supplemented with randomly sampled one hop neighbors as one example of such methods. However, such method may not be applicable to all KGs. We leave this for future works.

## G   Broader Impacts

In the real world, our culture, values and knowledge are always evolving. As one of the key basis for many down stream applications, knowledge graph should evolve accordingly as the new concepts emerge. However, such an update often incurs high costs from both manually adding in these new concepts as new entities and relations, as well as retraining all the downstream models. Our work provides a data efficient way to incorporate these new entities and relations to the existing knowledge graph, as well as an example model that dynamically incorporate new triplets during test time. Specifically across various scientific disciplines, our method can accelerate scientific discovery over the graph with human knowledge on chemistry, physics and biology, and provide justification and explanation why some facts about the new entities/relations are more promising than others. However, the imputed knowledge about the new entities and relations could be overly relied on and become misguidance. To mitigate this, the generated knowledge, especially ones based on only a few verified examples but have high stakes, should be verified by humans through domain-specific experiments.