# OpenReview forum: "Few-shot Relational Reasoning via Connection Subgraph Pretraining"
_NeurIPS.cc/2022/Conference — NeurIPS 2022 Accept_

### Official Review · Reviewer_LUSp · 2022-07-03

**Rating:** 6
**Confidence:** 3
**Soundness:** 2 fair
**Presentation:** 2 fair
**Contribution:** 3 good

**Summary:**

The author proposes  a method that can perform well on any novel few-shot tasks without relying on specifically designed meta-training set.
The proposed approach relies on an intuition that entities linked by the same relation should have a similar-strctured subgraph.
The framework is called connection subgraph reasoner, it has two major components: hypothesis proposal module and evidence proposal module. The hypothesis proposal module generates a hypothesis embedding from each support graph such that they are most similar to each other. Then, the evidence model attempts to find the closest evidence to the hypothesis embedding.
In a leanring-free optimization, the output masks are optimized directly.
There's also a learning version where the masks are outputed by a neural model.
In experiments strong performance is shown. Especially in inductive setting.

**Questions:**

Some part of the paper is not easy to follow. (see my questions)

**Limitations:**

I did not find a limitation section in the main text.

**Strengths And Weaknesses:**

The proposed method is novel.

I think the idea of subgraph matching is interesting.

In experiments strong performance is shown. Especially in  inductive setting.

---

> ### Author Response · Authors · 2022-07-26
> **Thank you for your review, missing question section?**
>
> Dear reviewer LUSp,
>
> Thank you for your review! You mentioned "Some part of the paper is not easy to follow. (see my questions)", however, we did not see them in the Questions section. May I know your questions to the paper? Thanks again for your review.
>
> Best,
> Paper8212 Authors

---

> > ### Comment · Reviewer_LUSp · 2022-08-08
> > **questions**
> >
> > Sorry, I forgot to attach the question.
> >
> > Question:
> >
> > Line63: In the folllowing example, "(cook, ./, kitchen) 63 example, such a connection subgraph that implies ./ is a two hop path in KG: {(chop, can be done 64 with, knife), (knife, is located at, kitchen)}." I don't see "cook"  in the two-hop path?
> >
> >
> > It's hard for me to follow equation 4 and 5. for example in equation 5, what is the max over? And what exactly is M_p? (you can say that you'll explain it later)

---

> > > ### Author Response · Authors · 2022-08-09
> > > **Thank you for the questions**
> > >
> > > Thank you for the questions and we apologize for the confusion from our writing. Please see our clarifications below:
> > >
> > > > “I don't see "cook" in the two-hop path?”
> > >
> > > Sorry that we mistyped “chop” as “cook” in Line 62 as also pointed out by another reviewer. This is supposed to be the same example as in the abstract and we will explicitly point this out in our future draft.
> > >
> > > >“It's hard for me to follow equation 4 and 5. for example in equation 5, what is the max over? And what exactly is M_p? (you can say that you'll explain it later)”
> > >
> > > In Equation 5, the max is over the masks m_i for all i, which will then be the output of M_p. Equation 5 defines what we intend M_p to be ideally, and we later define two specific implementations of M_p (Eq.12 - 14 and Algorithm 1). We will change Eq. 5 to using argmax similar to Eq. 12 to clarify this. Equation 4 shows how the hypothesis proposal module M_p is used in the hypothesis proposal stage in our framework (i.e. the upper half of (C) in Figure 1). Thank you for pointing out these clarifying issues of our notations and we will explicitly specify the variables of max and argmax. We will also explicitly explain that specific implementations of M_p will be explained later in our future draft.

---

> > > > ### Comment · Reviewer_LUSp · 2022-08-09
> > > > **thanks**
> > > >
> > > > Those are helpful, thanks!

---

> ### Author Response · Authors · 2022-08-02
> **Response**
>
> Thank you for acknowledging our contribution to the field of few-shot KG completion, the novelty of our method, and overall strong empirical performance. Due to the space limit, we included detailed discussions of the limitation of our method in the Appendix E. We will move it to the main text in the final version. We also plan to clarify several details and improve our writing as replies to other reviewers. Please let us know any additional questions you have and we are happy to answer them in this thread!

---

### Official Review · Reviewer_LgC2 · 2022-07-11

**Rating:** 7
**Confidence:** 3
**Soundness:** 3 good
**Presentation:** 3 good
**Contribution:** 3 good

**Summary:**

In this paper, the authors investigated few-shot knowledge graph completion task. Previous work usually model this under meta-learning framework. The authors proposed to leverage sub-graph between the head entity and tail entity, which may be shared between support and query triplets. Through this way, they are able to better predict the target entity through the shared subgraph, which contains richer local information that the model can leverage. To get a model that can be generalized to specific KG, they proposed to pretrained encoder-decoder model with the pretraining task to recover the masked nodes and relations in the subgraph. The experimental verify the effectiveness of the proposed model, which outperforms the baseline models on multiple existing datasets dramatically.

**Questions:**

What if there are no relationship between  subgraphs in support and query sets, will the pretrained model still work properly?

**Strengths And Weaknesses:**

Strengths:
* The authors proposed an approach to leverage local subgraph for few-shot relational reasoning tasks, which achieves state-of-the-art performance on multiple datasets.
* They proposed to pretrain an encoder-decoder model which can generalize to specific KG.
* The paper is well-written and easy to follow.

Weaknesses:
On some datasets like FB15K-237, the proposed approach underperforms an existing baseline model, which may raise the question of the generalization ability of the pretrained model and on what kind of datasets will this model performs better.

---

> ### Author Response · Authors · 2022-08-02
> **Response**
>
> Thank you for the very helpful feedback and comments! Below we address the concern and questions raised in the review:
>
> > “On some datasets like FB15K-237, the proposed approach underperforms an existing baseline model, which may raise the question of the generalization ability of the pretrained model and on what kind of datasets will this model performs better.”
>
> The reviewer states concerns about our model achieving a worse performance than the baseline on FB15k-237 and would like a further analysis on generalization based on this. Thank you for this suggestion. We note that we have already included some discussion in lines 280-283 about this. Our conjectured reason is that FB15k-237 is much denser than the other two knowledge graphs, so the pretrained entity embeddings are already very informative and predictive of the relational structure. In comparison, under the more realistic inductive setting where node embeddings may not be available, our method is able to significantly improve the predictive performance including on such a dense graph FB15k-237, with 20% improvement on MRR over the state-of-the-art baselines FSRL and MetaR (Table 3).
>
> > “What if there is no relationship between subgraphs in support and query sets, will the pretrained model still work properly?”
>
> This is a great question. Technically, our method does not assume any strict relationship between the subgraphs in the support set and query set, i.e. there does not need to be exactly the same common connection subgraphs or even the same types of edges. Instead, our method learns to find soft common connection subgraphs that have similar semantic meanings and implications. So we do have this mild assumption implicitly that such related subgraphs exist, since otherwise the support and query subgraphs are completely unrelated and the few shot problem is then probably not feasible.

---

### Official Review · Reviewer_iFPo · 2022-07-11

**Rating:** 7
**Confidence:** 3
**Soundness:** 3 good
**Presentation:** 2 fair
**Contribution:** 3 good

**Summary:**

* The paper presents a novel way of performing few-shot KB completion without having to rely on meta-train datasets constructed in an ad-hoc fashion. Specifically, they propose learning a hypothesis proposal module that given different support evidence graphs, finds a common hypothesis that is supported by the evidence. The hypothesis is then instantiated with the different examples and passed to the encoder to generate a representation of the support examples. During inference, the proposed method checks if there is an evidence close enough to the proposed hypothesis using the Evidence Proposal module to first generate a hypothesis, and then instantiating it with the query, and using the encoder to generate the query evidence representation.
* The authors present 2 approaches for the hypothesis proposal and evidence proposal modules: an optimization based training free method, and a fully trainable GCNN approach. For the latter, they present a self-supervised pre-training method for training.
* The optimization free method is competitive against other Meta learning approaches, while the fully trainable approach outperforms them, especially for the scenario where inference is performed on unseen entities (dubbed the inductive setting in the paper).

**Questions:**

Questions and Typographic improvements:
* Is there a hypothesis why even randomly initialized encoders work well with the training free approach. On a similar line of reasoning, would it be possible to get the results of using the training free approach, but leveraging the encoder trained with the learning based scheme ?
* For Algorithm 1, I think line 8 is incorrectly placed ? (It should be placed after line 6, if my understanding is correct)
* In Line 62, cook => chop ?
* For the Synthetic dataset experiment, it would be good to include some more details in the main paper from the appendix.

**Limitations:**

The authors have covered the limitations fairly well. I am a little concerned w.r.t inference cost of the proposed method. If it is indeed an issue, it would also be good to mention that in the limitations as well.

**Strengths And Weaknesses:**

Strengths:

* The paper presents an interesting approach for adapting to few-shot KB completion, which is contrary to the popular meta-learning approach.
* It is especially interesting to see that the training free approach, with randomly initialized encoders performs competitively against other Meta learning approaches.
 * The results especially for the transductive setting for ConceptNet and NELL, and inductive setting for all 3 datasets, are quite compelling with substantial improvements over the baseline. In addition, the approach also appears to be fairly robust to distribution shifts.

Weakness
* Given that during inference, for each entity pair, the model needs to compute a forward pass through Evidence Proposal Module and the Encoder Module, it would be good to also compare the inference time of the proposed method compared to the baseline approaches.
* Since the approach should produce interpretable masks, it might be helpful and insightful to include some qualitative examples. Eg: for the running example in the paper (chop, <>, kitchen), does the hypothesis proposal module actually identify the underlying accurate hypothesis, and is that correctly re-generated from the Evidence Proposal module.
* The definition of pre-training is a bit overloaded. From the Supplementary material, it seems like the model is jointly trained with all 3 losses, as opposed to first doing self-supervised pre-training followed by finetuning. It might be good to clarify that in the paper.
* [Minor] This setup for few-shot link prediction seems to be very well suited for in-context learning setups. I wonder if it makes sense to also compare against the same.
* [Minor] The paper is a bit hard to parse: some of the notation (eg: Equations 9 and 11) are somewhat non-standard, in Equation 16, T is not defined, and it is somewhat unclear how all the pieces fit together. It would be nice if the authors could present (even in the Appendix, if space is a concern) an Algorithm that clearly defines how the different proposals are connected for a training and inference pass through the dataset.

---

> ### Author Response · Authors · 2022-08-02
> **Response (1/2)**
>
> Thank you for the insightful comments and very helpful constructive feedback! Below we address a number of important points raised by your review.
>
>
> > “...It would be good to also compare the inference time of the proposed method compared to the baseline approaches.”
>
> We have measured the inference time of our method and baselines on NELL. Please find it in the table below. We find that our method has comparable inference runtime compared with state-of-the-art baselines FSRL but it’s slower than MetaR. The reason is that for each query triplet, our model needs to use the evidence proposal module to decode the edge masks and obtain the embeddings for the connection subgraphs. Compared with MetaR, which directly uses shallow KG embeddings to score each query triplet in TransE style, CSR achieves much better empirical performance and also allows inductive few-shot KG link prediction. We will add the table and discussion to the final version.
>
> | NELL (inductive)          | MetaR | FSRL  | CSR-GNN   |
> |----------------|-------|-------|-------|
> | Inference Time (s) | 5.52  | 14.24 | 17.50 |
> | MRR            | 0.355 | 0.180 | 0.511 |
>
> | NELL (transductive)          | MetaR | FSRL  | CSR-GNN   |
> |----------------|-------|-------|-------|
> | Inference Time (s) | 5.50  | 14.49 | 18.33 |
> | MRR            | 0.471 | 0.490 | 0.577 |
>
>
> > “...it might be helpful and insightful to include some qualitative examples.”
>
> This is a great suggestion to include some qualitative examples to demonstrate the interpretable edge-level masks/ connection subgraphs our model is able to discover. We further explored this and attach some visualizations of the connection subgraph discovered by our model in the NELL test set: https://drive.google.com/drive/folders/1gfECsiUJ0l5PUkZoIB9S9yRDfhUjDQI4?usp=sharing. Each figure visualizes the evidence connection subgraph discovered for the novel relation in the title. We included only relation names but not entity names to emphasize the topological similarity and reduce cluttering.
>
> In these figures, we can see that the model discovered meaningful evidence connection subgraphs to support its prediction, e.g. `concept:agriculturalproductgrowninlandscapefeatures, concept:geopoliticallocationcontainscountry => concept:agriculturalproductcamefromcountry`. The model also selects some one-hop neighbors of head/tail that are not reachable to the tail/head but provide information about the type of the head/tail entity: e.g. `concept:countrycities` helps to determine that the head entity is a country. Moreover, different subgraphs with similar semantics are identified when the exact same subgraph is not available,. We thank the reviewer for this suggestion and will add these visualizations and discussions in the final version.
>
>
> > “The definition of pre-training is a bit overloaded…”
>
> The reviewer points out that our usage of word “pretraining” is overloaded, since we use the two self-supervision losses together with a finetuning loss, instead of “first doing self-supervised pre-training followed by finetuning”. We would like to clarify that this “finetuning loss” is an end-to-end supervised pretraining loss over randomly sampled few-shot tasks in the pretraining data, instead of finetuning over meta-training tasks. We thank the reviewer for the clarification suggestion and will rename this loss to pretraining task loss to clarify this point.
>
>
> > “This setup for few-shot link prediction seems to be very well suited for in-context learning setups. I wonder if it makes sense to also compare against the same.”
>
> This is a great point. The few-shot link prediction setup can indeed be seen as a form of in-context learning, which is also one of the reasons why we are very interested in this problem. In this paper, we focus on the standard few-shot KG link prediction setting where we aim to infer the link using explicit knowledge graph structure but without textual information over the nodes/edges. Hence, directly comparing against language model in-context learning is out of the scope of this paper, but we definitely plan to further probe that direction in future works.
>
>
> > “Is there a hypothesis why even randomly initialized encoders work well with the training free approach.”
>
> The surprising performance with a training free encoder is a very interesting discovery to us and we are actively exploring this in our follow-up work. Our current hypothesis is that the random GNN encoder provides a good random feature since the random aggregation function hashes the multisets received by each node and edges well. This differentiable random hashing can distinguish different topologies well and provides meaningful gradients on this difference. One recent paper on arxiv presents similar findings along this line: https://arxiv.org/pdf/2201.05349.pdf. We intend to investigate this more in the future work.

---

> > ### Author Response · Authors · 2022-08-02
> > **Response (2/2)**
> >
> > > “would it be possible to get the results of using the training-free approach, but leveraging the encoder trained with the learning-based scheme?”
> >
> > This is an interesting suggestion. Empirically, we indeed noticed that using pretrained encoder with the learning-free method can increase MRR by about 1 point in NELL transductive setting. However, this seems unnecessary given that we can already obtain the pretrained GNN based model after pretraining encoder and decoder together. We will include this result in our appendix.
> >
> > > “The paper is a bit hard to parse…”
> >
> > Thank you for the various suggestions on writing. We will add a pseudo algorithm based on Algorithm 1 and 2 to show how the framework works end to end. We will also move line 8 to line 6 in Algorithm 1, replace “cook” with “chop” in line 62, and add more details on the synthetic experiments in the final version.

---

> > > ### Comment · Reviewer_iFPo · 2022-08-06
> > > **Acknowledgement of Author Response**
> > >
> > > Thank you for addressing the points of concerns. I do think the additions potentially improve the clarity from a reader's perspective. Overall, in my opinion, this is a solid work, and I retain my "Accept" recommendation.

---

### Author Response · Authors · 2022-08-02
**Thank you all for the helpful reviews**

We thank all the reviewers for the very helpful and insightful reviews! We are glad to see that all reviewers appreciate the value of our paper. Specifically, the reviewers (especially iFPo and LUSp) find the main idea of leveraging subgraph matching and pretraining for few-shot relational reasoning tasks novel and interesting. All reviewers agree that the proposed method provides strong and substantial empirical improvements. We also appreciate all the constructive feedback and questions. Please see our reply to each reviewer individually below.

---

### Meta-Review · Area_Chair_hEB5 · 2022-08-30

**Recommendation:** Accept
**Confidence:** Certain

**Metareview:**

This paper studies few-shot knowledge graph completion problem. It proposes learning a hypothesis proposal module that given different support evidence graphs, finds a common hypothesis that is supported by the evidence. The authors present 2 approaches for the hypothesis proposal and evidence proposal modules: an optimization-based training free method, and a fully trainable GCNN approach.

The reviewers agree that the proposed method is interesting and solid, the experiments are thorough, and the results provide valuable insights for future work. Reviewers' raised concerns and questions are properly addressed by the author's response.

**Award:**

No

---

### Decision · Program_Chairs · 2022-09-14

Accept